# Combining Transarterial Radioembolization (TARE) and CT-Guided High-Dose-Rate Interstitial Brachytherapy (CT-HDRBT): A Retrospective Analysis of Advanced Primary and Secondary Liver Tumor Treatment

**DOI:** 10.3390/cancers14010072

**Published:** 2021-12-24

**Authors:** Florian Nima Fleckenstein, Maximilian Julius Roesel, Maja Krajewska, Timo Alexander Auer, Federico Collettini, Tazio Maleitzke, Georg Böning, Giovanni Federico Torsello, Uli Fehrenbach, Bernhard Gebauer

**Affiliations:** 1Department of Diagnostic and Interventional Radiology, Charité-Universitätsmedizin Berlin, Corporate Member of Freie Universität Berlin and Humboldt-Universität zu Berlin, 13353 Berlin, Germany; maximilian.roesel@charite.de (M.J.R.); timo-alexander.auer@charite.de (T.A.A.); federico.collettini@charite.de (F.C.); georg.boening@charite.de (G.B.); giovanni-federico.torsello@charite.de (G.F.T.); uli.fehrenbach@charite.de (U.F.); bernhard.gebauer@charite.de (B.G.); 2BIH Charité Clinician Scientist Program, Berlin Institute of Health at Charité-Universitätsmedizin Berlin, 10178 Berlin, Germany; tazio.maleitzke@charite.de; 3Institute of Biometry and Clinical Epidemiology, Charité-Universitätsmedizin Berlin, Corporate Member of Freie Universität Berlin and Humboldt-Universität zu Berlin, 10117 Berlin, Germany; maja.krajewska@charite.de; 4Center for Musculoskeletal Surgery, Charité-Universitätsmedizin Berlin, Corporate Member of Freie Universität Berlin and Humboldt-Universität zu Berlin, 13353 Berlin, Germany; 5Julius Wolff Institute, Berlin Institute of Health at Charité-Universitätsmedizin Berlin, 13353 Berlin, Germany

**Keywords:** Interventional Radiology, Oncology, SIRT, Ablation, Minimal Invasive, Locoregional therapy

## Abstract

**Simple Summary:**

Clinical management of advanced stages of primary and secondary liver tumors remains challenging. Combining different treatment approaches to create the most effective therapy for patients is, however, often necessary. With this study we aim to analyze the efficacy and safety of a combined intrahepatic treatment of transarterial radioembolization and CT-guided high-dose-rate interstitial brachytherapy. Our study showed that patients not responding to systemic chemotherapy or suffering from tumor relapse after surgical resection might benefit from a combined minimal-invasive treatment.

**Abstract:**

Purpose: Treatment of patients with primary and secondary liver tumors remains challenging. This study analyzes the efficacy and safety of transarterial radioembolization (TARE) combined with CT-guided high-dose-rate interstitial brachytherapy (CT-HDRBT) for the treatment of primary and secondary liver tumors. Patients and Methods: A total of 77 patients (30 female) with various liver malignancies were treated. Primary endpoints were median overall survival (OS) and time to untreatable progression (TTUP). Additionally, subgroup analyses were performed in consideration of diagnosis and procedure sequence. Median OS and TTUP prediction were estimated using Kaplan–Meier analysis and hazard ratios (HR) were calculated using a multivariate Cox proportional hazard model. Results: A total of 115 CT-HDRBT and 96 TARE procedures were performed with no significant complications recorded. Median OS and TTUP were 29.8 (95% CI 18.1–41.4) and 23.8 (95% CI 9.6–37.9) months. Median OS for hepatocellular carcinoma (HCC)-, cholangiocarcinoma carcinoma (CCA) and colorectal cancer (CRC) patients was 29.8, 29.6 and 34.4 months. Patients starting with TARE had a median OS of 26.0 (95% CI 14.5–37.5) compared to 33.7 (95% CI 21.6–45.8) months for patients starting with CT-HDRBT. Hazard ratio of 1.094 per month was shown for patients starting with CT-HDRBT. Conclusion: Combining TARE and CT-HDRBT is effective and safe for the treatment of advanced stage primary and secondary liver tumors. Our data indicate that early TARE during the disease progression may have a positive effect on survival.

## 1. Introduction

Besides advancements in oncological therapies, the management of primary and secondary liver malignancies remains challenging. Liver metastases are often fatal, independently of their primary cancer, and the prognosis is poor [1]. Approximately one-third of all oncological patients suffer from metastases at the time of diagnosis, and 50% of patients diagnosed in early-stages subsequently develop metastases in the liver over the course of disease. Although incidences for primary liver cancer have decreased in the last three decades, they still remain high [2]. In 2018, liver cancer was found to be responsible for approximately 780,000 deaths worldwide, accounting for 8% of all cancer-related deaths [3]. For primary malignancies such as hepatocellular carcinoma (HCC) and cholangiocarcinoma carcinoma (CCA), surgical resection remains the therapy of choice but is often impossible due to inaccessibility, number of lesions and tumor distribution [4]. Furthermore, recurrence after resection is common [5,6]. Despite advances in systemic therapies, local treatment approaches using minimally invasive therapies (MIT) have proven to significantly prolong overall survival (OS) in patients with limited metastatic disease to the liver, supporting the concept of oligometastatic disease [7,8].

Transarterial radioembolization (TARE) showed good results of OS for both primary and secondary malignancies of the liver [9,10,11,12]. CT-guided high-dose-rate interstitial brachytherapy (CT-HDRBT) is an ablative technique by which a radioactive source (Iridium 192) is inserted into tumor lesions through catheters which have been implanted in the tumor under CT guidance [13,14]. It is being used by a growing number of centers around the world with excellent treatment results for the treatment of solid primary and secondary tumors [15,16,17,18]. In contrast to radiofrequency ablation (RFA) and microwave ablation (MWA), CT-HDRBT overcomes size limitations and restrictions due to tumor location (e.g., proximity to the liver hilum or vessels).

Both TARE and CT-HDRBT have been combined with a variety of MIT with good results regarding OS and safety [19,20]. Yet no study has evaluated the combination of both treatments. In this study, we evaluated the efficacy and safety of a combined treatment approach of TARE and CT-HDRBT in patients who received at least one TARE and one CT-HDRBT, regardless of diagnosis and pretreatment.

## 2. Methods

This study was performed in accordance with the standards of the Helsinki Declaration and was approved by the Charité ethical review board (EA4/08917) on 24 May 2017. Between March 2007 and November 2020, a total of 77 patients received at least one TARE and one CT-HDRBT. Written informed consent was obtained from each patient. Before all procedures a contrast enhanced Gd-EOB-DTPA (Primovist, Bayer, Leverkusen, Germany) MRI was acquired (Figure 1A,B). All indications for CT-HDRBT and TARE procedures were confirmed by a multidisciplinary tumor board. Demographics of all patients included are summarized in Table 1.

### 2.1. CT-HDRBT

CT-HDRBT is used in our institution for the treatment of unresectable liver only or dominant tumors or liver metastases. Criteria for performing the procedure are: (1) Liver function Child–Pugh Class A or B, (2) total bilirubin< 2 mg/dL, (3) platelet count >50,000/nL, (4) prothrombin time (PT) > 50%, and (5) partial thromboplastin time (PTT) < 50 s. If necessary, the haemostasis was improved. If present, ascites was drained before treatment to avoid bleeding. Exclusion criteria for CT-HDRBT include (1) any evidence of progressive extrahepatic tumor spread and (2) more than five intrahepatic tumor lesions. Of note, for CT-HDRBT there is no limit regarding the maximum size of a treated lesion [21].

All patients were treated under conscious sedation using midazolam and fentanyl. After local anaesthesia, the tumor lesion was punctured using a 17 G needle under CT-guidance. A flexible 6F angiographic sheath (Radiofocus™, Terumo, Japan) was then introduced into the hepatic target lesion over a stiff guide wire (Amplatz™, Boston Scientific, Boston, MA, USA) using the Seldinger technique. The guide wire was then removed and a closed-ended 6F afterloading catheter (Primed™, Halberstadt Medizintechnik GmbH, Halberstadt, Germany) was inserted through the sheath. Eventually, a CT scan of the liver was acquired to confirm correct catheter positions for three-dimensional radiation planning (Brachyvision, Varian Medical Systems, Palo Alto, CA, USA). Catheters, clinical target volume (CTV), and potential risk structures were plotted semi-automatically. Radiation target dose of the CTV was 20 Gy (Figure 1D). Maximum doses above 50 Gy were allowed in the tumor center. All irradiations were completed as single-fraction in afterloading technique using an Iridium-192 radiation source with a nominal activity of 370 Gbq. After irradiation, all catheters were carefully removed, and the puncture tracts sealed using thrombogenic sponge torpedoes (Gelfoam^®^ absorbable gelatin sponge, USP, Pfizer, New York, NY, USA) to minimize the risk of bleeding [15,22,23].

### 2.2. TARE

Generally, all liver tumors (both primary and metastatic) are potentially suitable for TARE and are generally considered for treatment if they fall into a subset that are (1) chemotherapy-refractory, (2) too advanced or technically not suitable for ablation and liver surgery. Patients are not treated if they show rapidly progressive extra-hepatic disease with no strategy available for an adequate disease control. At our institution the most common indication is third or subsequent line liver-only or liver-dominant chemotherapy-refractory metastatic colorectal cancer. General exclusion criteria are: (1) life expectancy >12 weeks, (2) ECOG/WHO performance status 0–2, and (3) adequate liver function (i.e., <bilirubin 34 µmol/L, i.e., 2.0 mg/dL).

TARE is a two-step procedure consisting of evaluation and therapy procedure, which has been described in detail previously [24,25,26]. Briefly, TARE evaluation contained an angiographic evaluation of the hepatic vasculature as well as, if needed, coil embolization of the gastroduodenal artery and the right gastric artery to prevent potential extrahepatic deposition of radioactive material. Subsequently, technetium-99 m labelled macroaggregated albumin acting as a surrogate marker was injected in the left and right hepatic artery. Afterwards, a single photon emission CT was performed to identify potentially extrahepatic uptake. Moreover, the CT scan serves as a tool to evaluate lung and gastrointestinal-tract shunt fractions. Approximately two weeks after evaluation, patients were prepared for treatment session. A Gd-EOB-DTPA (Primovist^®^, Bayer, Leverkusen, Germany) MRI was acquired to quantify liver, as well as tumor, volumes. Required dosage of resin Yttrium-90 (Y-90) microspheres (Sirtex, North Sydney, NSW, Australia) was calculated based on the dosimetric (partition) model [25]. After successful injection of the particles, a Y-90 PET/CT scan was acquired to determine the radiopharmaceutical distribution (Figure 1C).

### 2.3. Follow Up

Follow-up routine included MRI, clinical visits as well as a multidisciplinary tumor board case discussion. The MRI was obtained 6- and 12-weeks post-procedure before prolonging the interval to 3 months. Six months post-procedure chest imaging was included in the routine biannually. In case of stable disease or remission, this cycle was maintained for 18 months, before reducing MRI scans to biannual appointments (Figure 1E,F). MRI evaluation was performed by two board-certified radiologists in consensus. In case of tumor progression, all therapeutic approaches were performed in accordance with the multidisciplinary tumor board.

### 2.4. Endpoints and Statistical Analysis

Primary endpoints were median OS and time to untreatable progression (TTUP). TTUP was defined as the time from the first treatment with either TARE or CT-HDRBT to the exhaustion of all local therapy approaches [27]. Data collection ended in December 2020. Time of death was determined by using the internal hospital information system, searching for obituaries, and contacting general practitioners. Additionally, subgroup analyses were performed depending on the most frequent diagnoses and in consideration of the procedure sequences. Complications were classified according to the standards of the Society of Interventional Radiology [28]. The study design is graphically summarized using the PICOT format in Figure 2 [29].

Statistical analysis was conducted using SPSS (Statistical Package for the Social Sciences, version 27.0). Testing for normality was performed with the Shapiro–Wilk test. Normally-distributed continuous data were presented as the mean and standard deviation (SD) and non-normally distributed data were expressed as median and interquartile range (IQR) or range. Kaplan–Meier curves were used to analyse and visualize OS and TTUP. For the subgroup analysis, groups were compared using the nonparametric Mann–Whitney U test and Chi-squared test. Furthermore, a Cox regression model with time-dependent and time-independent covariates was used to analyze effects on survival. *p*-values of <0.05 were considered significant.

## 3. Results

The study population included 30 women and 47 men ranging in age from 22 to 85 years (median 63 years). The most common diagnoses were HCC (*n* = 37), colorectal carcinoma (CRC, *n* = 13), CCA (*n* = 9) and neuroendocrine tumor (NET, *n* = 9). Multiple patients underwent liver surgery (*n* = 20) as well as chemotherapy (*n* = 43), mostly before MIT. Furthermore, a majority of patients underwent other MITs such as transarterial chemoembolization (TACE) or RFA. Demographic characteristics are summarized in Table 1.

### 3.1. Procedures and Adverse Events

Our patients received a total of 115 CT-HDRBT treatments. Fifty-three patients were treated only once, fifteen were treated twice and nine patients had more than three treatments. A total of eight mild and two moderate adverse events were recorded: five patients developed free perihepatic fluid, two patients experienced nausea and one patient showed elevated temperatures post-treatment. All mild complications were treated pharmaceutically without any intervention needed. We recorded one moderate complication in a patient with a pneumothorax after CT-HDRBT, which was treated with a pleural drainage. The patient was discharged two days after treatment without any discomfort. A second patient with a moderate adverse event developed hyperbilirubinemia combined with severe pain and was therefore transferred to the department of gastroenterology where he was recompensated.

A total of 96 TARE therapies were performed. Fourteen patients underwent a sequential procedure with about six weeks in between treatments. Five patients received two TARE procedures. The mean activity delivered to the patient was 1.22 GBq (SD 0.54). Nine mild, one moderate and one severe AE occurred. The nine mild AE consisted of nausea (*n* = 8) and mild fever (*n* = 8) that were treated pharmacologically. One patient with a mild complication suffered from post-interventional ascites that was treated with an abdominal ascites drainage. No further intervention was necessary. The one patient with a severe AE suffered from a pseudoaneurysm at the puncture site of the femoral access, which was successfully treated using ultrasound-guided thrombin injection.

### 3.2. Primary Outcomes

The total survival rate after 12, 24 and 36 months was 85.7%, 56.2% and 39.8%, respectively (Figure 3). Median OS was 29.8 months (95% confidence interval [CI] 18.16–41.42). Median TTUP was 23.8 months (95% CI 9.61–37.93).

### 3.3. Subgroup Analysis

Baseline characteristics of our study subdivided depending on procedure sequence are summarized in Table 2. Patients starting with TARE showed a significantly shorter median duration before switching therapies than patients treated with CT-HDRBT first (*p* = 0.037). The total survival rate in the TARE before CT-HDRBT group after 12, 24 and 36 months was 94.7%, 51.7% and 39.4% with a median OS of 26.0 months. In contrast, for patients, who received CT-HDRBT before TARE survival rates were 81.9%, 58.0% and 40.2% after 12, 24 and 36 months with a median OS of 33.7 months. Log-Rank test could not be calculated since the proportional hazard assumption was violated (Figure 4A).

Therefore, we computed our stratification factor TARE before or after CT-HDRBT as a time-dependent covariate and included it in our Cox regression model. We additionally added TARE before or after CT-HDRBT as a time-independent covariate and the time between TARE and CT-HDRBT in our model. Per month passed, the risk of death increased by 1.094 (95% CI 1.027–1.165, *p* = 0.005) times for patients starting with CT-HDRBT compared to patients starting with TARE (time-dependent covariate). When not taking into account the time passed, no difference between the groups could be observed (time-independent covariate). Further, we could show that patients with less time between their therapies showed a lower OS with a Hazard Ratio of 0.922 (95%CI 0.889–0.956, *p* = 0.00001). Thus, the risk of death decreased almost 8% per month in between the two therapies.

Additionally, the cohort was stratified according to diagnoses with a median OS of 29.8, 29.6 and 34.4 months for HCC, CCA and CRC, respectively (Table 3). Of note, OS rate for NET did not drop below 65% (Figure 4) and therefore it was not possible to estimate the median OS for this subgroup. Survival rates after 12, 24 and 36 months for HCC patients were 88.5%, 53.6% and 39.6%.

## 4. Discussion

This study has three main findings. First, the concept of combining TARE with CT-HDRBT is an effective treatment for advanced-stage liver tumors. Second, it can be applied safely to a broad field of patients including extensive pretreatments and multiple tumor entities. Thirdly, our data indicate that an early TARE in the course of the disease might have a positive effect on survival.

The management of advanced stages of both primary and secondary liver tumors remains challenging and combining different approaches in order to create the most effective treatment for patients is often a clinical necessity. The lack of standardized treatments in these patients is met by a broad variety of minimal-invasive procedures [30]. The herein presented first analysis of a combined treatment of TARE and CT-HDRBT supports an individual combination of multiple minimal-invasive therapy approaches in order to maximize treatment success and the quality of patient care.

Merging therapeutic effects to maximize response to treatment is a clinical reality in the field of interventional radiology. Multiple studies report the efficacy and safety of combinations of minimal-invasive therapies such as TACE or TARE plus local ablative therapies such as RFA, MWA or CT-HDRBT.

For the treatment of advanced HCC, systemic chemotherapy can be distinguished into targeted therapies and immunotherapies. As a result of promising studies, tyrosine kinase inhibitors such as sorefenib and lenvatinib were granted approval in first-line therapy whereas checkpoint inhibitors such as nivolumab are considered as second-line therapy [31,32]. However, the efficacy of TKI is limited by the development of drug resistance. In this context, major neuronal isoform of (RAS)/Rapidly Accelerated Fibrosarcoma protein (RAF)/mitogen-activated and extracellular-signal regulated kinase (MEK)/extracellular-signal regulated kinases (ERK) pathways play a central role [33]. Hence, most large randomized multicenter studies showed disease control in only about 50% of cases, still lacking robust evidence for significant survival benefits. Moreover, especially treatment with tyrosine kinase inhibitors is known to be linked to severe limitations of quality of life [34]. In light of these caveats, therapies using local treatment approaches are today a clinical reality. A retrospective case control study with 240 patients showed an advantage in combining TACE with RFA compared to RFA alone [35]. In 2020, Wang et al. examined 183 patients with a recurrence of HCC, who were either treated with TACE alone or with RFA/MWA and TACE combined [36]. After propensity score matching, there were two groups including 65 patients each with no significant difference in their baseline characteristics. The TACE-Ablation group had 1, 3 and 5-year OS rates of 81.2%, 52.4%, 41.6% compared to the TACE-alone group with only 64.9%, 36.6%, 30.2%, showing a clear advantage for combined treatments. A previously conducted study examined 47 patients, who were treated with TACE or TAE combined with CT-HDRBT [19]. The TAE group achieved a median OS of 32.3 months and the TACE group 28.9 months. The 37 HCC patients in our cohort achieved a median OS of 29.8 months, confirming these results.

Regarding CCA, the recently presented MISPHEC Trial was conducted in seven centers in France and included 41 patients with unresectable CCA [37]. The first-line treatment encompassed chemotherapy (gemcitabine + cisplatin) and TARE in combination. This prospective study showed a median OS of 22 months.

CT-HDRBT in the context of CCA has been investigated in multiple studies in the past [38,39,40]. In a recently published study, 61 CCA patients received 96 CT-HDRBTs in total [38]. The study reported a median OS of 15.5 months for lesions smaller than 4 cm (*n* = 18) and 10.0 months for larger lesions (*n* = 43). We can report a median OS of 29 months for CCA patients indicating a very good response to a combined therapy of TARE and CT-HDRBT.

Regarding CRC metastases, a conducted study evaluated 23 patients who were treated with TACE using Irinotecan-loaded microspheres and CT-HDRBT in combination [41]. The authors highlighted the overall safety and good feasibility of this procedure and reported median OS of eight months. TARE is known for being safe, which we can confirm by only recording one AE in 15 sessions for 13 patients. A recently published study examined 131 patients with CRC metastases and showed a median OS of 10.7 months [42]. We can report a median OS of 34 months for our CRC patient subgroup indicating very good response to treatment in our cohort.

A retrospective study evaluating the efficacy of TARE on NET metastatic to the liver analysed 40 patients treated with 56 sessions in total [43]. The authors report a median OS of 24.7 months. Survival rates of patients with NET metastatic to the liver treated with CT-HDRBT is reported with a 5-year OS rate of 63% [44]. In spite of the small patient numbers in this patient subgroup, results indicate that a combination of TARE and CT-HDRBT seems reasonable, especially since our nine patients’ total survival rate remained above 65%.

Nine patients with liver metastases from other origins than previously described were also included in our study. We did not notice any abnormalities in safety and feasibility.

The Cox regression model shows a significant survival benefit for patients treated with TARE early in the course of disease. Hence, we assume that TARE sufficiently stabilises tumor progression. This result stands in contrast to the clinical reality, where most patients receive TARE rather late in the course of disease. Even though, it is not infrequently observed that an earlier TARE shows good results in both short- and long-term outcomes [42,43,45].

The present study demonstrates that TARE and CT-HDRBT can effectively be combined in patients suffering from advanced primary and secondary liver tumors. According to current guidelines, most of the patients in our cohort would have only qualified for best supportive care since they were non-responders to chemotherapy or suffered from tumor relapse after surgical resection. Yet, treating these patients by deviating from current guidelines is a clinical reality and this study therefore addresses a topic that might be of interest beyond the field of interventional radiology. This unique combination merges an unselective, whole liver approach of TARE with a focused and high-dose approach of CT-HDRBT. With combining both treatment principles, we are able to treat advanced tumors of various origins successfully with excellent outcomes regarding median OS, TTUP and no significant complications. We believe that with the development of new targeted chemo- and immunotherapies the need for a combination of treatment strategies will emerge in the very near future. In this context the present study focusses on one potential element of future therapies designed for patients suffering from cancer in advanced stages.

This study has several limitations. Firstly, it has a retrospective single-center design. This might limit transferability to other oncological centers. Moreover, while the heterogeneity of our study population provided previously mentioned advantages, it nevertheless might weaken comparability. Statistical calculations were furthermore limited by small numbers within the analysed subgroups. The results of this study are not based on a robust statistical dataset, especially in tumors that are not commonly treated using TARE and ablation such as metastases from pancreatic or breast cancer. However, since a combination of TARE with any other MIT is generally rarely performed, it is unlikely to find a much larger patient cohort. In order to confirm the findings of this study statistically powered clinical trials will be necessary in the future.

## 5. Conclusions

A combination of TARE and CT-HDRBT offers an effective and safe treatment approach for a broad range of advanced primary and secondary liver malignancies. The promising median OS and TTUP presented in this study are encouraging regarding the use of different treatment combinations according to the individual course of diseases. Finally, the herein presented results indicate that a treatment with TARE early in the course of disease might be beneficial with regard to survival outcomes.

## Figures and Tables

**Figure 1 cancers-14-00072-f001:**
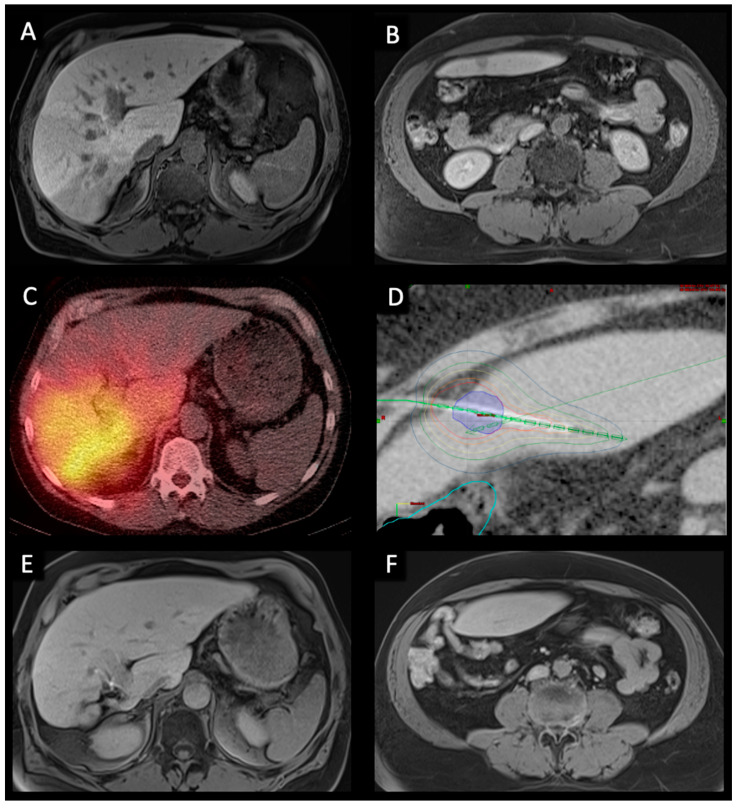
Pre-, peri-, and post-interventional imaging of a 56-year-old male with bilobar HCC treated with right lobar TARE and left lobar CT-HDRBT at intervals of six weeks. (**A**,**B**). Pre-interventional transversal contrast-enhanced MRI in the hepatobiliary excretion phase showing a large infiltrative HCC in segment VI/VII as well as a smaller contralateral metastasis in segment III. (**C**). Post-TARE PET/CT scan showing good radiopharmaceutical distribution of Y-90 spheres in the right liver. (**D**). CT-HDRBT peri-interventional 3D-irradiation plan using contrast-enhanced CT after CT-guided positioning of the afterloading catheter. Visible tumor borders were defined as the clinical target volume (CTV) (blue area). Dose distribution was adjusted by 3D-treatment planning. The planned minimal enclosing dose was 20 Gy. Isodose irradiation lines surround the CTV. The colon was marked (light blue line) to minimize collateral radiation. (**E**,**F**). 36-months post-interventional transversal contrast-enhanced MRI examination in the hepatobiliary excretion phase showing complete response to treatment as well as hypertrophy of the right liver lobe.

**Figure 2 cancers-14-00072-f002:**
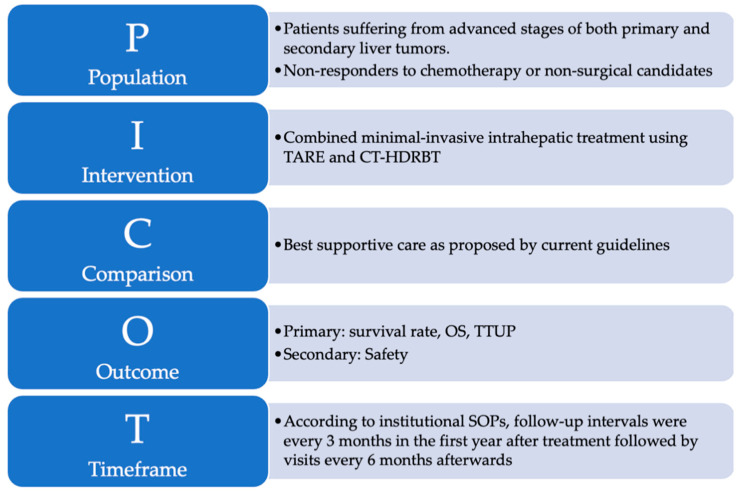
The PICOT format is a helpful and reader-friendly approach for summarizing research questions that explore the effect of treatment interventions: (P)–Population refers to the sample of subjects in this study. (I)–Intervention refers to the treatment that was provided to subjects enrolled in this study. (C)–Comparison identifies the reference group of patients to compare with the treatment intervention. (O)–Outcome represents the outcome parameters of this study. (T)–Time describes the duration of data collection.

**Figure 3 cancers-14-00072-f003:**
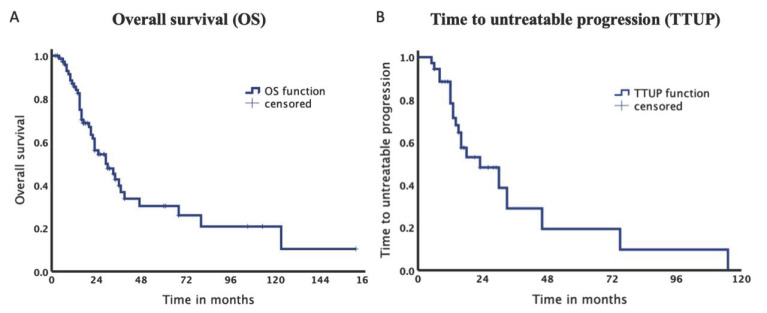
Kaplan–Meier Curves of the entire collective. (**A**). Median overall survival was 29.8 months. (**B**). Median time to untreatable progression was 23.8 months.

**Figure 4 cancers-14-00072-f004:**
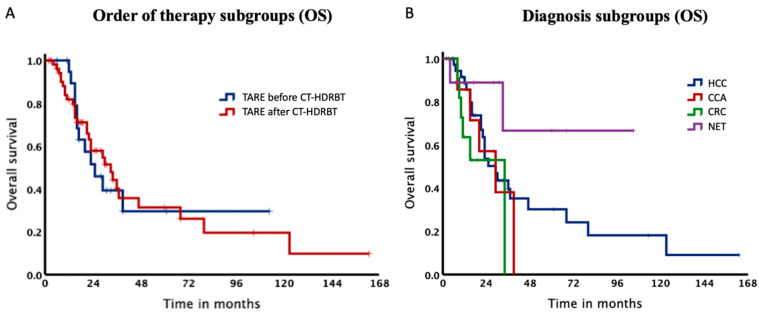
Kaplan–Meier Curves of the subgroups. (**A**). Median overall survival was 26.0 months for patients, who received TARE before CT-HDRBT and 33.7 months, when receiving TARE after CT-HDRBT. (**B**). Kaplan–Meier Curves of our subgroups. Median overall survival for HCC, CCA and CRC was 29.8, 29.6 and 34.4 months.

**Table 1 cancers-14-00072-t001:** Patient cohort characteristics.

Number of Patients	77	CT-HDRBT before TARE (%)	55 (71.4)
Median age, years (range)	63 (22–85)	TARE before CT-HDRBT (%)	22 (28.6)
Sex (female/male)	30/47	Number of CT-HDRBT per patient (%)	
Diagnosis (%)		1	53 (68.8)
HCC	37 (48.1)	2	15 (19.5)
CCA	9 (11.7)	3+	9 (11.7)
CRC	13 (16.9)	Number of CT-HDRBT in total	115
NET	9 (11.7)	Adverse events CT-HDRBT ^b^ (%)	10 (8.7)
Uveal melanoma	4 (5.2)	1	8 (7.0)
Breast cancer	2 (2.6)	2	2 (1.7)
Pancreatic cancer	2 (2.6)	Number of TARE per patient (%)	
Cervical cancer	1 (1.3)	1	58 (75.3)
Primary/Liver Metastasis	46/31	2	19 (24.7)
Liver surgery (%)	20 (26.0)	Sequential procedure ^c^ (%)	14 (73.7)
Before	19 (95.0)	Number of TARE in total	96
After	1 (5.0)	Adverse events TARE ^b^ (%)	11 (11.5)
Patients with other MIT ^a^ (%)	39 (50.6)	1	9 (9.5)
Chemotherapy (%)	43 (55.8)	2	1 (1.0)
Before	35 (81.4)	3	1 (1.0)
Between	5 (11.6)	TARE locus (%)	
After	3 (7.0)	Whole liver	44 (57.1)
Median duration (months) from diagnosis to first CT-HDRBT/TARE (IQR)	14.5 (3.4–39.4)	Right liver	20 (26.0)
		Left liver	13 (16.9)
Median duration (months) between first CT-HDRBT and first TARE (IQR)	9.2 (4.1–21.0)		
Mean TARE dose, GBq (SD)	1.22 (0.54)		

^a^ Minimally invasive therapy: TACE (28), TACE + TAE (1); TACE + RFA (2); RFA (2); PRRT (3); PRRT + TAE (2); cryotherapy (1). ^b^ Adverse Event Classification by the Society of Interventional Radiology. ^c^ 14 of the 19 patients with two TARE therapies had a sequential procedure. Treatment was split in two sessions, starting with one liver lobe and approximately 6–8 weeks later the contra-lateral lobe. These patients were counted as whole liver treatment. Abbreviation: HCC Hepatocellular carcinoma; CCA Cholangiocarcinoma; CRC Colorectal Cancer; NET Neuroendocrine Tumor; CT-HDRBT Computed Tomography-Guided High-Dose-Rate Interstitial Brachytherapy; TARE Transarterial Radioembolization; RFA Radiofrequency ablation; TACE Transarterial chemoembolization; TAE Transarterial embolization; PRRT Peptide receptor radionuclide therapy; IQR Interquartile range; SD Standard derivation; m month.

**Table 2 cancers-14-00072-t002:** Subgroup characteristics regarding procedure sequence.

	TARE after CT-HDRBT	TARE before CT-HDRBT	*p*-Value
Number of patients	55	22	
Median age, years (range)	63 (22–85)	64 (36–79)	0.624 ^a^
Sex (male/female)	37/18	10/12	0.076
HCC + CCA/Metastasis	32/23	14/8	0.659
Liver surgery (%)	14 (25.5)	6 (27.3)	0.869
Patients with further MIT	30 (54.5)	9 (40.9)	0.28
Chemotherapy	29 (52.7)	14 (63.6)	0.384
Number of CT-HDRBT per Patient			0.525 ^a^
1	37 (67.3)	16 (72.7)	
2	10 (18.2)	5 (22.7)	
3+	8 (14.5)	1 (4.6)	
Adverse events CT-HDRBT	7 (8.2)	3 (10.0)	>0.05
Number of TARE per patient			0.740 ^a^
1	42 (76.4)	16 (72.7)	
2	13 (23.6)	6 (27.3)	
Adverse events TARE	8 (11.8)	3 (10.7)	>0.05
Median duration (m) from diagnosis until first CT-HDRBT/TARE (IQR)	14.5 (3.2–37.6)	14.3 (3.3–43.7)	0.795 ^a^
Median duration (m) between first CT-HDRBT and first TARE (IQR)	12.6 (5.1–25.0)	8.0 (2.9–11.4)	0.037 ^a^

^a^ *p*-values are calculated using Mann–Whitney U test. Remaining *p*-value are calculated using Chi-squared test. Values are given as *n* (%) or median (range or interquartile range). Abbreviation: HCC Hepatocellular carcinoma; CCA Cholangiocarcinoma; CRC Colorectal Cancer; NET Neuroendocrine Tumor; CT-HDRBT Computed Tomography-Guided High-Dose-Rate Interstitial Brachytherapy; TARE Transarterial Radioembolization; IQR Interquartile range; m month.

**Table 3 cancers-14-00072-t003:** Subgroup patient characteristics according to diagnoses.

	HCC	CCA	CRC(LM)	NET(LM)
Number of patients	37	9	13	9
Median age, years (range)	67 (54–85)	61 (43–85)	56 (48–59)	63 (36–77)
Gender (male/female)	27/10	5/4	9/4	4/5
Liver surgery	8 (21.6)	3 (33.3)	3 (23.1)	6 (66.7)
Patients with further MIT	23 (62.1)	2 (22.2)	5 (38.5)	6 (66.7)
Chemotherapy	11 (29.7)	7 (77.8)	12 (92.3)	8 (88.9)
Number of CT-HDRBT per patient				
1	23 (62.1)	7 (77.8)	9 (69.2)	7 (77.8)
2	7 (18.9)	1 (11.1)	4 (30.8)	2 (22.2)
3 +	7 (18.9)	1 (11.1)	0 (0.0)	0 (0.0)
Adverse events CT-HDRBT	4 (6.5)	0 (0.0)	4 (23.5)	1 (9.1)
Number of TARE per patient				
1	30 (81.1)	7 (77.8)	11 (84.6)	6 (66.7)
2	7 (18.9)	2 (22.2)	2 (15.4)	3 (33.3)
Adverse events TARE	6 (13.6)	0 (0.0)	1 (6.7)	2 (16.7)
TARE before CT-HDRBT	10 (27.0)	4 (44.4)	1 (7.7)	3 (33.3)
Median duration (m) from diagnosis until first CT-HDRBT/TARE (IQR)	3.9 (2.3–9.4)	14.4 (6.8–31.9)	28.4 (14.0–38.3)	52.4 (36.7–131.6)
Median duration (m) between first CT-HDRBT and first TARE (IQR)	12.4 (4.0–26.2)	11.0 (3.0–21.3)	7.0 (5.1–13.6)	18.0 (4.8–59.3)

Values are given as *n* (%) or median (range or interquartile range). Abbreviation: HCC Hepatocellular carcinoma; CCA Cholangiocarcinoma; CRC Colorectal Cancer; NET Neuroendocrine Tumor; CT-HDRBT CT-Guided High-Dose-Rate Interstitial Brachytherapy; TARE Transarterial Radioembolization; IQR Interquartile range; LM Liver metastases; m month.

## Data Availability

The data presented in this study are available on request from the corresponding author. The data are not publicly available due to ethical restrictions.

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
