# Peer review of "Combining Transarterial Radioembolization (TARE) and CT-Guided High-Dose-Rate Interstitial Brachytherapy (CT-HDRBT): A Retrospective Analysis of Advanced Primary and Secondary Liver Tumor Treatment"

_cancers, 2021, doi:10.3390/cancers14010072_

Round 1

Reviewer 1 Report

Florian Nima Fleckenstein et al. uncovered the efficacy and safety of selective internal radiation therapy (SIRT) combined with CT-guided high-dose-rate interstitial brachytherapy (CT-HDRBT) for the treatment of primary and secondary liver tumors.

Point to be addressed:

  1. I would suggest adding a short summary figure (graphical abstract or similar) inspired to the following points:

    please consider the PICOT format as a helpful and reader-friendly approach for summarizing research questions that explore the effect of the authors' intervention:

    • (P) – Population refers to the sample of subjects you wish to recruit for your study. There may be a fine balance between defining a sample that is most likely to respond to your intervention (e.g. no co-morbidity) and one that can be generalized to patients that are likely to be seen in actual practice.

    • (I) – Intervention refers to the treatment that will be provided to subjects enrolled in your study.

    • (C) – Comparison identifies what you plan on using as a reference group to compare with your treatment intervention. Many study designs refer to this as the control group. If an existing treatment is considered the ‘gold standard’, then this should be the comparison group.

    • (O) – Outcome represents what result you plan on measuring to examine the effectiveness of your intervention. Familiar and validated outcome measurement tools relevant to common chiropractic patient populations may include the Neck Disability Index or Roland-Morris Questionnaire.There are, typically, a multitude of outcome tools available for different clinical populations, each having strengths and weaknesses.

    • (T) – Time describes the duration of your data collection.

    • 2. The study design is not perfectly clear and I would suggest slightly emphasizing the trial limitations (need to statistically powered trial aiming to corroborate the authors' hypothesis-generating findings, especially in terms of survival). I would tune down the prognostic impact in light of this observation. 

    • 3. This reviewer personally misses some important biological background that can boost the interest for a broad readership from the oncology field focusing on the authors' findings: sorafenib pretreatment is known to enhance radiosensitivity through targeting MAPK in HCC, nonetheless,  for several years, sorafenib, as a tyrosine kinase inhibitors (TKI) inhibitor, has been the approved treatment option, to date, for advanced HCC patients. Its activity is the inhibition of the retrovirus-associated DNA sequences protein (RAS)/Rapidly Accelerated Fibrosarcoma protein (RAF)/mitogen-activated and extracellular-signal regulated kinase (MEK)/extracellular-signal regulated kinases (ERK) signaling pathway. However, the efficacy of sorafenib is limited by the development of drug resistance, and the major neuronal isoform of RAF, BRAF and MEK pathways play a critical and central role in HCC escape from TKIs activity. Advanced HCC patients with a BRAF mutation display a multifocal and/or more aggressive behavior with resistance to TKI (please refer to PMID: 31766556 and expand introduction/discussion sections)

Reviewer 2 Report

Dear Editor, thank you so much for inviting me to revise this manuscript.

This study addresses a current topic.

The manuscript is quite well written and organized. English could be improved.

Figures and tables are comprehensive and clear.

The introduction explains in a clear and coherent manner the background of this study.

We suggest the following modifications:

  • Introduction section: although the authors correctly included important papers in this setting, we believe a couple of studies should be cited within the introduction ( PMID: 30993888 ;  PMID: 33571059), only for a matter of consistency. We think it might be useful to introduce the topic of this interesting study.
  • Methods and Statistical Analysis: nothing to add.
  • Discussion section: Very interesting and timely discussion. Of note, the authors should expand the Discussion section, including a more personal perspective to reflect on. For example, they could answer the following questions – in order to facilitate the understanding of this complex topic to readers: what potential does this study hold? What are the knowledge gaps and how do researchers tackle them? How do you see this area unfolding in the next 5 years? We think it would be extremely interesting for the readers.

However, we think the authors should be acknowledged for their work. In fact, they correctly addressed an important topic, the methods sound good and their discussion is well balanced.

One additional little flaw: the authors could better explain the limitations of their work, in the last part of the Discussion.

We believe this article is suitable for publication in the journal although some revisions are needed. The main strengths of this paper are that it addresses an interesting and very timely question and provides a clear answer, with some limitations.

We suggest a linguistic revision and the addition of some references for a matter of consistency. Moreover, the authors should better clarify some points.

Reviewer 3 Report

This manuscript was an original article which retrospectively investigated the efficacy and safety of selective internal radiation therapy (SIRT) combined with CT-guided high-dose-rate interstitial brachytherapy (CT-HDRBT) for the treatment of primary and secondary liver tumors. The authors demonstrated that SIRT with CT-HDRBT was an effective treatment for advanced stage liver tumors. Furthermore, they indicated that an earlier SIRT in the course of disease might have a positive effect on survival.

This study was conducted well, and the methods used are appropriate. The data are presented clearly. In general, this is a well-written paper that presents interesting data. This article will be of interest to clinicians and researchers involved in this field. The following minor issue should be addressed:

Minor

  1. (Title) Please insert “high-dose-rate” in “CT-guided interstitial brachytherapy”.
  2. (Figure 1) The cross-section level in Figure 1F seems different from one in Figure 1B.
  3. I recommend that the authors described adverse events in detail, which is required to reveal safety of this treatment.
  4. Please describe the indication of this treatment in the Method section.
  5. There are several treatments, including operation, RFA, TACE and chemotherapy for advanced liver tumor. The authors should discuss the expected position of SIRT with CT-HDRBT in the treatment strategy.

Round 2

Reviewer 1 Report

The authors have clarified several of the questions I raised in my previous review. Most of the major problems have been addressed by this revision.

Reviewer 2 Report

The authors modified the manuscript according to our suggestions.

We recommend Acceptance.